A machine learning approach to managing game bird introductions

Smith Austin M. ams89@ufl.edu 1 2
Cropper, Jr Wendell P. 3
Moulton Michael P. 2
1 School of Natural Resource and Environment, University of Florida , Gainesville , FL , United States of America
2 Department of Wildlife Ecology and Conservation, University of Florida , Gainesville , FL , United States of America
3 School of Forest, Fisheries, and Geomatics Sciences, University of Florida , Gainesville , FL , United States of America
Garant Dany
Electronic publication date: 2025 Nov 4
Publication date: 2025
Volume: 13
Electronic Location ID: e20291
Received 2025 Jul 3; Accepted 2025 Oct 3
Copyright: ©2025 Smith et al.
Copyright year: 2025
Copyright holder: Smith et al.
License: This is an open access article distributed under the terms of the Creative Commons Attribution License, which permits unrestricted use, distribution, reproduction and adaptation in any medium and for any purpose provided that it is properly attributed. For attribution, the original author(s), title, publication source (PeerJ) and either DOI or URL of the article must be cited.
License URL: https://creativecommons.org/licenses/by/4.0/

Keywords: Alectoris chukar, Ensemble modeling, Wildlife management, Habitat suitability, Species distribution modeling, Species introductions

Funding: The authors received no funding for this work.

==============================
Effective management of introduced species requires a clear understanding of their habitat requirements. Species distribution models (SDMs) offer a powerful tool for addressing this challenge. We applied seven modeling techniques to predict a suitable habitat for the introduced Chukar Partridge (Alectoris chukar), including artificial neural networks, generalized additive models, k-nearest neighbor, random forests, support vector machines, extreme gradient boosting, and a weighted ensemble approach. Using site-level data on physiography, climate, land cover, and habitat range, we modeled Chukar distributions by simulating historical introduction efforts and extrapolating predictions into surrounding areas to assess cross-regional transferability. Model performance was evaluated using independent, geographically distinct validation datasets. Our results demonstrate that machine learning-based SDMs provide accurate and transferable predictions of Chukar habitat suitability. This study highlights the value of machine learning for predicting establishment success while emphasizing the importance of incorporating species movement behavior and site fidelity into SDM frameworks. Overall, our findings contribute to advancing conservation planning, species reintroductions, and adaptive management strategies.

Introduction

The large-scale release of captive-bred species for introduction is a common management practice, especially for species with commercial or recreational value (Bilal, 2022; Larsen et al., 2007; Lever, 2005; Long, 1981; Moulton et al., 2018; Parish & Sotherton, 2007). Many countries established formal acclimatization programs during the 19th and 20th centuries, aiming to enrich local fauna, enhance hunting opportunities, or provide food resources through the introduction of non-native species (Lever, 2005; Long, 1981). Throughout the 20th century, the Foreign Game Investigation Program (FGIP)—a United States (U.S.) government-initiative—focused on identifying, acquiring, and evaluating non-native wildlife species for potential introduction, primarily as game animals (Bohl & Bump, 1970; Bump, 1941; Bump, 1951; Bump, 1963; Moulton et al., 2018; Smith, Cropper Jr & Moulton, 2021). This program aimed to supplement native game populations by introducing species with favorable traits (e.g., hardiness, high reproductive potential, and hunting appeal) into areas where native species were scarce or declining. This process often involved experimental releases, habitat assessments, and extensive field trials. Several avian species, particularly within the family Phasianidae, were evaluated and released as part of this initiative (Banks, 1981; Bohl, 1957a; Bohl, 1957b; Bohl & Bump, 1970; Bump, 1941; Bump, 1963; Bump, 1968; Gullion, 1965).

One of the most notable successes was the introduction of the Chukar Partridge (Alectoris chukar). Chukars are ground-dwelling Eurasian Galliformes that favor high-elevation, arid habitats dominated by talus slopes, sparse grasses, and shrub cover (Alcorn & Richardson, 1951; Barnett, 1952; Bohl, 1957a; Christensen, 1970; Christensen, 2020; Galbreath & Moreland, 1953; Gruychev, Dyakov & Dimitrov, 2014; Harper, Harry & Bailey, 1958; Smith, Cropper Jr & Moulton, 2021). They are among the most widely introduced gamebirds worldwide, with both notable successes and failures. In the conterminous U.S., introductions were attempted in at least 42 states, though self-sustaining populations are now confined to ten western states. Their core distribution lies in the Great Basin, extending into eastern Washington, northern Idaho, western Wyoming and Colorado, the northwestern corner of Arizona, and parts of Montana (Christensen, 1970; Christensen, 2020). Successful introductions also occurred in the Hawaiian Islands, New Zealand, and through translocations within the native range, whereas efforts in Australia, western Europe, and southern Africa were largely unsuccessful (Lever, 2005; Long, 1981).

Although programs like the FGIP helped expand gamebird diversity in some regions, they have been criticized for emphasizing a few successful introductions—such as the Chukar—while overlooking many failed attempts where species did not establish breeding populations (Gullion, 1965; Pierce, 1956). Success is often attributed to factors like large release numbers (e.g., Blackburn, Lockwood & Cassey, 2009; Blackburn, Lockwood & Cassey, 2015; Blackburn et al., 2013; Lockwood, Cassey & Blackburn, 2005), as seen in states like Utah (185,911), Oregon (113,675), and Washington (50,900); however, many of these large propagules commonly involved releases after initial introductions had succeeded (Barnett, 1952; Harper, Harry & Bailey, 1958; Moulton & Cropper, 2015; Moulton & Cropper, 2016). Chukars were not uniformly distributed across these states, often traveling significant distances (i.e., 5–50 km) to reach preferred habitats, even when release sites appeared similar (Bohl, 1957a; Galbreath & Moreland, 1953). Despite large-scale releases, several attempts in states including Minnesota (85,000), Wisconsin (43,013), and New Mexico (31,000) failed to establish populations, raising concerns about the effectiveness of large-scale introductions (Moulton & Cropper, 2015; Moulton & Cropper, 2016; Moulton & Cropper, 2019; Moulton et al., 2018). These inconsistencies highlight that large propagule sizes alone cannot guarantee success, and that environmental variability plays a critical role (Moulton & Cropper, 2015; Moulton & Cropper, 2016; Smith, Cropper Jr & Moulton, 2021; Smyth & Drake, 2022; van der Marel, Waterman & López-Darias, 2021). Even so, traditional game management largely depended on broad, qualitative habitat evaluations and trial-and-error introduction strategies (Christensen, 1970; Gullion, 1965; Lever, 2005; Long, 1981; Pierce, 1956), which lacked precision and often ignored important local factors. This not only led to ineffective outcomes but also raised financial and ethical concerns due to high mortality in unsuitable habitats (Bilal, 2022; Gullion, 1965; Madden, Santilli & Whiteside, 2020). Thus, a key challenge moving forward is identifying locations with ideal environmental conditions that truly drive success while minimizing unnecessary losses.

Recent advances in computing technology and comprehensive data sets, particularly over the last two decades, have enabled the use of more sophisticated statistical and quantitative methods to better identify suitable habitats and assess trade-offs between environmental variables (Beery et al., 2021; Elith et al., 2006; Franklin, 2010; Guisan, Thuiller & Zimmermann, 2017; Guisan et al., 2013; Howard et al., 2014; Peterson et al., 2011; Zimmermann et al., 2010). For example, species distribution models (SDMs), can address these limitations by incorporating detailed data and statistical analysis to predict the potential geographic range of a species based on its known locations and habitat characteristics (Austin, 2002; Elith & Leathwick, 2009; Guisan & Thuiller, 2005; Guisan et al., 2013; Miller, 2010; Norberg et al., 2019; Valavi et al., 2022). SDMs are routinely built using advanced algorithms known as machine learning, a type of artificial intelligence that enables computers to learn from data and improve their performance over time without being explicitly programmed to mechanistically predict (Beery et al., 2021; Crisci, Ghattas & Perera, 2012; Olden, Lawler & Poff, 2008; Pichler & Hartig, 2023; Ramampiandra et al., 2023; Zhang & Li, 2017). These algorithms identify patterns in data, make predictions, or classify information based on known samples (Kuhn & Johnson, 2013).

SDMs are commonly used to assess the impacts of future climate change (e.g., Austin & Van Niel, 2011; Pearson & Dawson, 2003) and to map potential invasion corridors after a species has established (e.g., Barbet-Massin et al., 2018; Gallien et al., 2012; Mainali et al., 2015); however, they are less frequently employed to inform intentional species introductions for game management . To address this gap, Smith, Cropper Jr & Moulton (2021) assessed the predictive power of SDMs in identifying suitable sites for Chukar introductions by examining how site-level environmental factors relate to establishment success across the contiguous U.S., using ensemble models based on data from the species’ native range. While their results underscored the role of local environmental conditions in shaping outcomes, the models primarily predicted broad potential ranges rather than identifying precise introduction sites. Moreover, the authors only considered locations with known occurrences, excluding areas without observations. To improve predictive accuracy and inform future efforts, it is essential to compare successful and failed introduction sites—an approach that can help isolate key environmental drivers of establishment and guide more effective introductions.

In this study, we explore the use of machine learning-based SDMs as a practical tool to guide intentional gamebird introductions. Using the Chukar Partridge as a case study, we analyze historical introduction outcomes in Washington—one of the few events with records documenting both successes and failures across a range of perceived suitable and unsuitable habitats. Our first objective was to use SDMs to predict outcomes in select regions and then apply the models across the rest of the state. Our second goal was to evaluate the ability of these models to generalize beyond the training region by testing predictions in neighboring Oregon. By comparing results across both states, we assess model performance under varied outcomes and refine our understanding of the environmental drivers behind successful introductions. Ultimately, this approach aims to improve the precision and efficiency of future introductions, moving away from traditional trial-and-error strategies toward more informed and cost-effective practices.

Materials & Methods

Study area and context

We simulated the Chukar introduction efforts in Washington State, as documented by Barnett (1952) and Galbreath & Moreland (1953) (Fig. 1). Washington is one of the few states where both successful and unsuccessful Chukar releases have been recorded, in areas perceived as both suitable and unsuitable. The state’s environmental diversity—from its wet western regions to its arid eastern landscapes—offers a valuable gradient for examining the factors that influence establishment success (Alcorn & Richardson, 1951; Barnett, 1952; Galbreath & Moreland, 1953; Gohain, 1959). Accordingly, we used the Washington counties where Chukars were introduced as the foundation for our models.

Figure 1 Map of Chukar Partridge (Alectoris chukar) introduction outcomes across Washington State.

County-level outcomes are based on Barnett (1952) and Galbreath & Moreland (1953). Counties are shaded according to establishment results: blue indicates successful introductions, orange indicates failed introductions, and gray indicates inconclusive outcomes. Yellow points show GBIF occurrence records used for model training and validation. Maps were generated in R using Natural Earth polygon data (https://www.naturalearthdata.com), GBIF records (https://doi.org/10.15468/dl.5ybphp), and county identifications based on Barnett (1952), Galbreath & Moreland (1953), and Washington Department of Fish and Wildlife (2014–2023) (https://wdfw.wa.gov/hunting/management/game-harvest).

Between 1938 and 1951, over 5,800 Chukars were released across 24 counties. The most consistent success occurred in the drier, eastern counties, which closely resemble the species’ native range (Barnett, 1952; Galbreath & Moreland, 1953; Lever, 2005; Long, 1981). Other attempts failed, likely due to unsuitable land cover. Releases on the western side of the Cascades failed quickly—largely attributed to the region’s damp climate. In eight counties, mainly located in the southeastern quadrant, outcomes were inconclusive.

Following the eventual success in Washington, and given its similar environmental conditions, Oregon initiated its own widespread Chukar release program (Lever, 2005; Long, 1981). Unlike Washington, Oregon focused exclusively on areas deemed environmentally suitable, releasing approximately 50,000 Chukars between 1951 and 1956. By 1967, the population had grown substantially, with an estimated 1,235,000 individuals distributed across the state.

To evaluate factors influencing Chukar establishment success, we first developed SDMs based on introduction records from Washington State. Our initial focus was on counties with variable introduction outcomes, which served as a testing ground to assess the predictive capacity of habitat suitability modeling under uncertain establishment conditions. Using presence records from counties with confirmed successful introductions as the training dataset, we calibrated the model to identify key environmental conditions associated with establishment success and extrapolated habitat suitability predictions across all of Washington to assess broader spatial patterns.

To replicate the approach used by the Oregon Game Commission—whose site selection strategy was based on perceived habitat similarity—we refined our Washington-based model for application in Oregon. Specifically, models were trained using the entire Washington dataset, incorporating both training and testing records, to maximize the representation of environmental variation within that state. By capitalizing on shared environmental features and ecological analogs between Washington and Oregon, we projected habitat suitability across Oregon using these Washington-trained models alone. This approach enabled us to evaluate the generality of SDMs across state boundaries and to critically assess the ecological soundness of Oregon’s management decisions based on habitat similarity.

Data collection

Species records

We obtained Chukar occurrence records using data from the Global Biodiversity Information Facility (GBIF.org, 2025), accessed via the R package rgbif (Chamberlain et al., 2022). To improve data accuracy, we used the CoordinateCleaner package (Zizka et al., 2019) to identify and exclude records with imprecise or erroneous coordinates, as well as remove duplicate occurrences. It is important to note that, although Chukars possess a well-established naturalized range, numerous occurrence records fall outside of this area. These outliers are typically attributable to releases from private game ranches or escaped domestic individuals, which are generally not recognized as part of self-sustaining naturalized populations (Christensen, 2020). To focus on biologically relevant populations, we limited our dataset to records from counties in both Washington (Washington Department of Fish Wildlife, 2014-2023; Table S1) and Oregon (Oregon Department of Fish Wildlife, 2014–2023; Fig. S1; Table S2) that reported annual hunting harvests of Chukars. We also filtered GBIF records to match the temporal scope of harvest data (2014–2023). This resulted in 1,634 occurrences from Washington and 2,676 from Oregon.

For modeling purposes, we partitioned occurrence data into four distinct subsets: (1) a training dataset comprising records from Washington counties with confirmed establishment success; (2) a test dataset including all remaining Washington occurrences not used for model training; (3) a reduced test dataset containing records from counties where introduction outcomes were classified as inconclusive; and (4) an external test dataset consisting exclusively of occurrence records from Oregon.

Pseudo-absence sampling

Most biodiversity data sources, such as GBIF, are presence-only—they indicate where a species has been observed, but not where it was surveyed and not found (i.e., absent). As a result, species distribution models (SDMs) often rely on pseudo-absences: data points drawn from areas lacking recorded occurrences and assumed to represent less suitable conditions (Barbet-Massin et al., 2012; Elith & Leathwick, 2009; Phillips et al., 2009; Zbinden et al., 2024). While several methods exist for selecting pseudo-absences, each comes with caveats (Barbet-Massin et al., 2012; Senay, Worner & Ikeda, 2013). The most basic approach involves randomly sampling locations across the study area, excluding known presences. Although easy to implement, this method risks selecting areas that are environmentally suitable but unoccupied or under-sampled. To improve ecological relevance, environmental filtering restricts pseudo-absence selection to locations with environmental conditions that differ substantially from those associated with known presences, helping the model better distinguish between suitable and unsuitable habitat (Barbet-Massin et al., 2012; Lobo, Jiménez-Valverde & Hortal, 2010; Zbinden et al., 2024). Similarly, geographic constraints impose spatial buffers—such as excluding areas within a certain distance of known presences—to reduce the likelihood of selecting false absences near potentially suitable areas (Senay, Worner & Ikeda, 2013; Van Der Wal et al., 2009).

In this study, we employed a combined approach using both environmental filtering and geographic constraints. For model training, pseudo-absences were drawn from both counties with recorded failed and successful introductions. A 5 km buffer was applied around each occurrence point, reflecting the approximate minimum daily movement observed in Chukars and accounting for potential observation error or uncertainty (Galbreath & Moreland, 1953). To ensure a balanced training dataset, we matched the number of pseudo-absences to the number of presence records (Barbet-Massin et al., 2012). For model testing, we sampled an additional 10,000 pseudo-absences from counties not included in the training set and combined them with occurrence data outside of the training counties. Likewise, for testing model predictions in Oregon, 10,000 pseudo-absences were sampled across the state, excluding areas within a 5 km buffer of known occurrences.

Environmental data

All environmental data were obtained using the geodata package in R, which provides access to various global geospatial datasets, specifically designed for environmental and ecological modeling. The spatial datasets were in raster format with a 1km resolution and included: WorldClim bioclimatic covariates (Fick & Hijmans, 2017), a set of 19 variables summarizing climate variability over a 30-year period, with mean values calculated for each quadrangle; the European Space Agency’s (ESA) land cover classification, which consists of ten categories representing the proportion of each land cover type per raster pixel (Zanaga et al., 2021); and the NASA Shuttle Radar Topography Mission (SRTM) elevation raster layer, which provides data to calculate elevation, slope, aspect, and the Terrain Roughness Index (http://srtm.csi.cgiar.org/). Finally, because Chukars are were frequently observed near rivers and other water sources (e.g., Bohl, 1957a; Christensen, 1970; Galbreath & Moreland, 1953; Harper, Harry & Bailey, 1958), we calculated the distance to the nearest water body for each pixel using spatial polygon data from the rnaturalearth package in R (Massicotte & South, 2025).

In total, we compiled 33 potential covariates for use in modeling (Table 1). To ensure equal weighting among covariates, all raster layers were normalized to a 0–1 scale prior to point-based extraction and any subsequent preprocessing steps (Han, Pei & Tong, 2022; Smith, Cropper Jr & Moulton, 2021). Additionally, we addressed potential issues of multicollinearity and high dimensionality—known to impair statistical and machine learning model performance—by reducing the number of predictors and retaining only those that were relatively uncorrelated (Fourcade, Besnard & Secondi, 2018; Hastie, Tibshirani & Friedman, 2009; James et al., 2013; Kuhn & Johnson, 2013; Valavi et al., 2022). Accordingly, we used a correlation matrix to identify and exclude highly correlated variables, retaining only those with absolute Pearson correlation value below 0.8. This process reduced our set of input covariates from 37 to 17, retaining all ten landcover classes, four bioclimatic variables, measured elevation, and calculated slope and distant from water.

Table 1 Summary of measured environmental covariates.

Note: Covariates shown in bold font have an absolute Pearson correlation value below 0.8 and were retained for model construction.

Variable	Source	
BIO1 – Annual mean temperature (°C)	WorldClim	
BIO2 – Mean diurnal range (mean of monthly (max temp - min temp)) (°C)	WorldClim	
BIO3 – Isothermality (BIO2/BIO7) (×100) (°C)	WorldClim	
BIO4 – Temperature seasonality (standard deviation ×100) (°C)	WorldClim	
BIO5 – Max temperature of warmest month (°C)	WorldClim	
BIO6 – Min temperature of coldest month (°C)	WorldClim	
BIO7 – Temperature annual range (BIO5-BIO6) (°C)	WorldClim	
BIO8 – Mean temperature of wettest quarter (°C)	WorldClim	
BIO9 – Mean temperature of driest quarter (°C)	WorldClim	
BIO 10 – Mean temperature of warmest quarter (°C)	WorldClim	
BIO 11 – Mean temperature of coldest quarter (°C)	WorldClim	
BIO 12 – Annual precipitation (mm)	WorldClim	
BIO 13 – Precipitation of wettest month (mm)	WorldClim	
BIO 14 – Precipitation of driest month (mm)	WorldClim	
BIO 15 – Precipitation seasonality (coefficient of variation) (mm)	WorldClim	
BIO 16 – Precipitation of wettest quarter (mm)	WorldClim	
BIO 17 – Precipitation of Driest Quarter (mm)	WorldClim	
BIO 18 – Precipitation of warmest quarter (mm)	WorldClim	
BIO 19 – Precipitation of coldest quarter (mm)	WorldClim	
Tree cover (%)	ESA	
Shrubland (%)	ESA	
Grassland (%)	ESA	
Cropland (%)	ESA	
Built-up/urban (%)	ESA	
Bare/sparse vegetation (%)	ESA	
Snow and ice (%)	ESA	
Permeant water body (%)	ESA	
Herbaceous wetland (%)	ESA	
Moss and lichen (%)	ESA	
Elevation (m)	SRTM	
Slope (rad)	Calculated	
Terrain Roughness Index	Calculated	
Distance from water source (m)	Calculated	

Statistical methods

Model building framework

We used a supervised learning procedure for our model building; that is, models were trained to fit our input variables to a known response variable (e.g., habitat suitability). We used the ‘caret’ package (Kuhn, 2008; Kuhn & Johnson, 2013), an all-in-one platform that helps streamline machine learning modeling procedures, for our model analysis. ‘caret’ is a useful tool for non-expert practitioners as it automates the model building process by generating a series of models with different hyperparameter combinations, and then choses the best model based on an internal testing statistic. In our framework, we applied the 5-fold cross-validation which was performed five times resulting in 25 training samples. This process identifies the optimal hyperparameters and subsequently retrains the model on the entire training dataset using the selected parameters, which is then ready to be used for predictions.

We employed six widely used machine learning algorithms—commonly applied in ecological and geospatial modeling—to classify potential Chukar habitat distributions. These included artificial neural networks (ANN; Lek & Guégan, 1999), K-nearest neighbors (KNN; Chirici et al., 2016; Franco-Lopez, Ek & Bauer, 2001), generalized additive models (GAM; Guisan, Edwards Jr & Hastie, 2002), Random Forest (RF; Breiman, 2001; Valavi et al., 2021), support vector machines (SVM; Drake, Randin & Guisan, 2006), and extreme gradient boosting (XGBoost; Chen & Guestrin, 2016; Valavi et al., 2022), recommended over the conventional gradient boosting machine (i.e., GBM; De’Ath, 2007; Elith, Leathwick & Hastie, 2008; Friedman, 2001). A summary of each algorithm is provided in Table 2. To minimize individual model bias and enhance prediction reliability, we also generated an ensemble model based on the averaged outputs of all six algorithms (Araújo & New, 2007; Friedman, 2001; Kaky et al., 2020; Thuiller et al., 2009).

Table 2 Summary of algorithms with their respective R package extension.

Method	Overview	R library	
ANN	Created to replicate the human brain, these models use several stacked, fully connected layers of information-processing units (i.e., ‘neurons’) that transform input data into more manageable features for processing. Each neuron incorporates an activation function (e.g., sigmoid function) to decide if the processed information is important for the model’s learning. Neurons are connected through weighted scalars, which determine the strength of the connections and are recalibrated throughout the model training phase.	nnet	
GAM	An extension of generalized linear models that allows for nonlinear relationships between predictors and response variable. Models are the effect of each predictor using smooth functions (i.e., splines) instead of linear functions and the prediction is the sum of the individual effects of each variable. Models are optimized using component wise boosting—a process where each sequential iteration corrects the learning errors of the previously attempt.	mboost	
KNN	A non-parametric method based on input data mapping. Models store and visualize all training data in a multidimensional feature space, with each point labeled by its class. New samples are classified based on the majority vote of the nearest points when projected into the feature space.	base	
RF	An ensemble method that generates a series of fully grown, unpruned decision trees, constructed from bagging (i.e., bootstrap aggregation). Each tree is provided different bootstrapped samples of the data and a random subset of features at each decision node. The final decision (i.e., model output) is the averaged score (regression) or the majority vote (classification) amongst all trees.	randomForest	
SVM	A supervised method that maps input data, where models use a hyperplane to achieve the maximum separation between output classes. The hyperplane is positioned by the closest neighboring points (i.e., support vectors) to maximize the distance between them and the decision boundary.	e1071	
XGBoost	An ensemble method that generates a series of weak predictive decision trees, built from a subset of available input variables, and calibrates hyperparameters using gradient boosting—a process where each sequential tree corrects the learning errors of the previously trained tree(s). The models incorporate regularization techniques, which enhance generalization and reduce overfitting.	mboost	

Model evaluation metrics

All models were initially evaluated using the area under the receiver operating characteristic curve (AUROC) a widely used metric that summarizes the trade-off between sensitivity and specificity across a range of classification thresholds (Hastie, Tibshirani & Friedman, 2009; James et al., 2013; Kuhn & Johnson, 2013). Sensitivity refers to the proportion of true positives (i.e., correctly predicted species presences) among all actual presences, while specificity is the proportion of true negatives (i.e., correctly predicted absences) among all actual absences. AUROC values range from 0 to 1, with 0.5 indicating performance no better than random chance. Models with AUROC >0.7 are generally considered useful, while values above 0.9 are considered excellent; values below 0.5 suggest confusion between classes and unreliable predictions (Guisan, Thuiller & Zimmermann, 2017).

Because species distribution models (SDMs) typically generate continuous suitability scores (e.g., ranging from 0 to 1), they are informative for ecological interpretation but less intuitive for decision-making (Liu et al., 2005; Liu, Newell & White, 2016). To simplify model evaluation and facilitate interpretation in applied contexts, we converted continuous predictions into binary classifications (suitable vs. unsuitable) using the optimized specificity–sensitivity threshold (Barbet-Massin et al., 2012; Liu, Newell & White, 2016). This threshold was selected to maximize the true skill statistic (TSS), calculated as sensitivity plus specificity minus one. TSS evaluates the model’s ability to correctly distinguish between presences and absences and, importantly, is not influenced by class prevalence. In general, TSS values above 0.6 are considered to indicate good model performance (e.g., González-Ferreras, Barquín & Peñas, 2016).

Results

Model performance

Models performed strongest when evaluating samples from across the entire Washington testing regions (Table 3). The ensemble model achieved the highest score, with an AUROC of 0.98 and TSS scores of 0.8. This was followed closely by RF and XGBoost, both yielding AUROC = 0.97 and TSS = 0.84. ANN and GAM showed similar performances with AUROC values of 0.97 and TSS values of 0.82. KNN also produced competitive results (AUROC = 0.96, TSS = 0.83), while SVM showed the lowest performance (AUROC = 0.94, TSS = 0.75).

Table 3 Model prediction statistics for Washington testing regions and Oregon.

Accuracy metrics are area under the receiver operating characteristic (AUROC) curve and the true skill statistic (TSS).

	Washington (all testing regions)	Washington (unsure regions)	Oregon	
Model	AUROC	TSS	AUROC	TSS	AUROC	TSS	
Ensemble	0.98	0.85	0.92	0.71	0.90	0.62	
ANN	0.97	0.82	0.87	0.66	0.86	0.57	
GAM	0.97	0.82	0.88	0.65	0.88	0.58	
KNN	0.96	0.83	0.90	0.66	0.79	0.48	
RF	0.97	0.84	0.91	0.69	0.91	0.64	
SVM	0.94	0.75	0.77	0.45	0.85	0.64	
XGBoost	0.97	0.84	0.91	0.68	0.87	0.55	

In assessing Washington’s unsure regions, performance declined across all models (Table 3). The ensemble (AUROC = 0.92, TSS = 0.71), RF (AUROC = 0.91, TSS = 0.69) and XGBoost (AUROC = 0.91, TSS = 0.68) maintained top-tier performance. KNN also performed well in this subset (AUROC = 0.90, TSS = 0.66), while GAM (AUROC = 0.88, TSS = 0.65) and ANN (AUROC = 0.87, TSS = 0.66) both demonstrated moderate performance. SVM exhibited a notable decline, producing the lowest TSS (0.45) and an AUROC of 0.77.

In Oregon, model performance for predicting Chukar habitat suitability varied across algorithms but overall demonstrated strong predictive ability. RF and the ensemble model achieved the highest performance, each with an AUROC of 0.90 and TSS values of 0.62 and 0.61, respectively. SVM and XGBoost followed, both with TSS values of 0.57; however, SVM had a lower AUROC of 0.83 compared to 0.87 for XGBoost. GAM and ANN produced slightly weaker results, with GAM achieving an AUROC of 0.86 and a TSS of 0.54, and ANN an AUROC of 0.83 and a TSS of 0.51. Although both fall just below the 0.6 TSS threshold for “good” classification, they still reflect moderate predictive performance. KNN exhibited the weakest results, with the lowest AUROC (0.76) and TSS (0.43). While still exceeding the AUROC threshold for utility, KNN did not meet the TSS criterion for reliable classification, indicating relatively limited predictive capacity in this context.

Spatial suitability predictions

Suitability maps for Washington, generated by each modeling algorithm, are presented in Fig. 2 and compared against known species occurrence points. Predicted suitability is ranked on a continuous scale from 0 to 1, with higher values indicating greater predicted suitability. Across all models, higher suitability was generally concentrated in the central and eastern regions of the state. Among the models, the Ensemble, RF, and XGBoost exhibited the strongest spatial agreement, producing sharply defined high-suitability areas that closely aligned with observed occurrence data. ANN and GAM also captured the overall distribution of suitable areas but showed less spatial precision. In contrast, KNN and SVM showed lower spatial specificity and produced broader and more diffuse suitability patterns.

Figure 2 Predicted spatial distribution of Chukar Partridge (Alectoris chukar) across Washington.

The leftmost panel shows observed occurrence records used to train and evaluate the models. Remaining panels show predicted habitat suitability from the ensemble model and six individual algorithms: artificial neural networks (ANN), generalized additive models (GAM), random forests (RF), support vector machines (SVMs), k-nearest neighbors (KNN), and extreme gradient boosting (XGBoost). Suitability values range from 0 (dark) to 1 (light), with each model capturing different spatial patterns and prediction intensities.

Suitability predictions for Oregon were notably less consistent across models (Fig. 3). In general, the western portion of the state was predominantly predicted as low suitability, which aligns with the absence of observed occurrences in that region. The Ensemble, RF, and XGBoost models again demonstrated the most structured and spatially focused predictions, showing strong correspondence with known presence locations. Predictions from ANN and GAM were similar in pattern but tended to overpredict suitable areas beyond known occurrence zones. KNN highlighted areas of high suitability but frequently overgeneralized, suggesting potential overfitting in spatial extrapolation. The SVM model produced the most conservative prediction, identifying only a few isolated zones of high suitability near dense occurrence records.

Figure 3 Predicted spatial distribution of Chukar Partridge (Alectoris chukar) across Oregon.

The leftmost panel shows observed occurrence records used to train and evaluate the models. Remaining panels show predicted habitat suitability from the ensemble model and six individual algorithms: artificial neural networks (ANN), generalized additive models (GAM), random forests (RF), support vector machines (SVM), k-nearest neighbors (KNN), and extreme gradient boosting (XGBoost). Suitability values range from 0 (dark) to 1 (light), with each model capturing different spatial patterns and prediction intensities.

Discussion

This study highlights the utility of machine learning-based SDMs as a robust framework for informing intentional species introductions in game management. By simulating historical introduction efforts of the Chukar Partridge in Washington and projecting predictions into Oregon, we evaluated the capacity of SDMs to identify environmentally suitable habitats and assess their transferability across state boundaries. Overall, our models achieved strong performance, particularly the ensemble, RF, and XGBoost algorithms, which consistently showed high classification accuracy and spatial concordance with known species occurrences.

Importantly, our results highlight the benefits of ensemble modeling. A common challenge in SDMs is selecting an appropriate algorithm, particularly in the context of hyperparameter tuning and avoiding overfitting (Araújo & New, 2007; Guisan, Thuiller & Zimmermann, 2017; Norberg et al., 2019; Valavi et al., 2022; Zhang & Li, 2017). Ensemble approaches or model averaging offer robust solutions to these challenges (Araújo & New, 2007; Guisan, Thuiller & Zimmermann, 2017; Smith, Cropper Jr & Moulton, 2021; Thuiller et al., 2009). In our study, the ensemble model exhibited the highest spatial specificity, with sharply defined high-suitability zones concentrated near known presence points. This level of spatial precision is especially valuable for applied decision-making, allowing managers to prioritize candidate introduction sites with greater confidence.

Notably, individual algorithms can also be highly effective (Hao et al., 2020; Kaky et al., 2020; Norberg et al., 2019; Valavi et al., 2022). Both the RF and XGBoost models performed exceptionally well, which is not surprising given that these are themselves ensemble-based decision tree models—further reinforcing the strength of ensemble learning approaches. At the same time, non-ensemble methods such as ANN and GAM produced promising predictions, consistent with previous studies supporting their use in ecological modeling (Franklin, 2010; Guisan, Edwards Jr & Hastie, 2002; Lek & Guégan, 1999). In contrast, models such as SVM and KNN, while statistically sound, generated broader and more diffuse suitability patterns. These results may reflect challenges associated with high-dimensional feature spaces and sensitivity to local density variations (i.e., curse of dimensionality; Hastie, Tibshirani & Friedman, 2009; James et al., 2013), making such approaches less suitable for fine-scale ecological modeling.

The ability of the Washington-trained models to generalize to Oregon also provides insights into the ecological transferability of SDMs. Despite differences in data origin, the top-performing models demonstrated strong predictive power when applied to Oregon’s introduction landscape. This suggests that shared environmental gradients can facilitate the successful application of models trained in one region to another, especially when care is taken to ensure ecological and climatic congruence (Araújo & Peterson, 2012; Barbet-Massin et al., 2012; Van Der Wal et al., 2009; Velazco et al., 2024). The fact that Oregon’s large-scale Chukar introductions were concentrated in regions our models predicted as highly suitable offers retrospective support for the state’s management strategy—despite its original reliance on qualitative assessments—and further asserts the critical role of habitat conditions in introduction success.

At the same time, model accuracy declined when applied to counties in Washington with historically variable introduction outcomes. This reduction in performance likely reflects the ecological ambiguity of these transitional zones, which lie near the threshold of environmental suitability. However, these areas are particularly valuable for evaluating model resolution and sensitivity, as they capture subtle environmental gradients that may influence establishment outcomes. Notably, the continued strong performance of several models—such as the ensemble, Random Forest, and XGBoost—even in these uncertain contexts, suggests that species distribution models can still provide meaningful insights under ecologically ambiguous scenarios.

Nonetheless, several limitations warrant consideration. As with all modeling frameworks, it is important to choose covariates that are ecologically informative, but also linearly independent to reduce overfitting (Duan et al., 2014; Hastie, Tibshirani & Friedman, 2009; James et al., 2013; Norberg et al., 2019). Undoubtably, all covariates measured in the study can affect local populations; nonetheless, this information was highly correlated and/or redundant, which could lead to model bias (Araújo & Peterson, 2012; Hastie, Tibshirani & Friedman, 2009; James et al., 2013; Strobl et al., 2007). Here we reduced the dimensionality by reducing multicollinearity; however, it is possible the incorporation of other variables or different selection methods (e.g., different correlation thresholds, clustering, replacement, etc.) could improve model performance (Austin & Van Niel, 2011). Additionally, our environmental predictors—though comprehensive—may still omit key variables related to microhabitat conditions, interspecies competition, or anthropogenic influences (e.g., land use changes). Beyond abiotic factors, studies have shown that behavioral dynamics, local species interactions, and biotic events (e.g., grazing pressure, vegetation shifts, or predator abundance) can strongly influence species distribution and productivity (Gruychev, Dyakov & Dimitrov, 2014; Pearson & Dawson, 2003; Wittmann et al., 2016). Finally, our models rely on historical occurrence records, which may be influenced by observer bias or inconsistent survey effort (Phillips et al., 2009; Zizka et al., 2019).

Similarly, pseudo-absence selection requires careful consideration to avoid mischaracterization of habitat suitability (Barbet-Massin et al., 2012; Phillips et al., 2009). From a methodological standpoint, our inclusion of pseudo-absences derived from both successful and failed introduction counties enhances the ecological realism of our models. This approach enabled direct comparisons between suitable and unsuitable environments, rather than relying solely on presence-only data,which are often spatially biased or clustered (Elith & Leathwick, 2009; Peterson et al., 2011; Van Der Wal et al., 2009). To minimize the risk of incorporating false absences and improve model discrimination, we applied environmental filtering and spatial buffering which previous studies have shown improve model predictions (e.g., Barbet-Massin et al., 2012; Phillips et al., 2009; Senay, Worner & Ikeda, 2013; Van Der Wal et al., 2009; Zbinden et al., 2024). However, some areas classified as unsuitable may have been inaccurately represented. For example, Chukars released from captivity are highly mobile and generally exhibit low site fidelity, complicating efforts to accurately infer habitat associations (Bohl, 1957a; Christensen, 1970; Christensen, 2020; Galbreath & Moreland, 1953; Harper, Harry & Bailey, 1958). Additionally, individuals have been documented dispersing more 50 km from introduction points (Bohl, 1957a; Galbreath & Moreland, 1953; Harper, Harry & Bailey, 1958)—well beyond the buffering thresholds used in our analyses. Such mobility may influence model performance, particularly when predictions are extended to regions of uncertain establishment (Barbet-Massin et al., 2018; Velazco et al., 2024). We therefore recommend that future studies critically evaluate assumptions regarding species movement and site fidelity when selecting pseudo-absences, as these methodological choices can substantially affect model outputs.

Despite these challenges, the modeling framework presented here offers a significant advancement over traditional introduction strategies. By combining high-resolution environmental data, robust machine learning algorithms, and empirical records of both success and failure, this approach allows wildlife managers to make more informed and efficient decisions. As species introductions become increasingly subject to ecological, ethical, and economic scrutiny, the integration of SDMs into environmental planning represents a critical step toward effective management practices.

Conclusion

Our study demonstrates that SDMs, especially with machine learning ensemble methods and biologically relevant pseudo-absence selection, can predict gamebird establishment successfully across diverse landscapes. Additionally, our models provided strong predictions for retrospective evaluation and future management planning. The consistency of model performance across different algorithms reinforces the importance of habitat suitability in introduction success, while also highlighting the need for careful consideration of species movement behavior and site fidelity in SDM construction. These findings highlight the effectiveness of machine learning and ensemble modeling in guiding species introductions, reintroductions, and broader conservation strategies.

Supplemental Information

Supplemental Information 1 Historical records of chukar releases and hunting harvests

Supplemental Information 2 Hunting harvest statistics for chukar partridge (Alectoris chukar) in Washington State (2014–2023)

Supplemental Information 3 R scripts for data acquisition, preprocessing, model development, and spatial prediction generation

Supplemental Information 4 Hunting harvest statistics for chukar partridge (Alectoris chukar) in Oregon State (2014–2023)

Supplemental Information 5 Oregon map displaying county-level wildlife management boundaries

Maps were created by Austin M. Smith using R with Natural Earth polygon files (https://www.naturalearthdata.com), GBIF records (https://doi.org/10.15468/dl.5ybphp), and the records identifying the counties were based on Galbreath & Moreland (1953) and the Oregon Department of Fish and Wildlife. 2014–2023 (https://myodfw.com/articles/upland-birds-harvest-information).

The authors acknowledge the use of AI-based language assistance, Copilot (Microsoft) and ChatGPT (OpenAI), to assist with drafting tasks such as text editing, language refinement, rephrasing and summarization. All AI-generated content was reviewed and revised by the authors to ensure accuracy and scientific integrity. All scientific analyses, interpretations, and conclusions are the authors’ original work.

Additional Information and Declarations

Competing Interests

Author Contributions

Data Availability

The authors declare there are no competing interests.

Austin M. Smith conceived and designed the experiments, performed the experiments, analyzed the data, prepared figures and/or tables, authored or reviewed drafts of the article, and approved the final draft.

Wendell P. Cropper, Jr conceived and designed the experiments, performed the experiments, analyzed the data, authored or reviewed drafts of the article, and approved the final draft.

Michael P. Moulton conceived and designed the experiments, performed the experiments, analyzed the data, authored or reviewed drafts of the article, and approved the final draft.

The following information was supplied regarding data availability:

The data and the R code are available in the Supplemental Files.

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
