# Peer review of "A machine learning approach to managing game bird introductions"

_PeerJ, doi:10.7717/peerj.20291_

## Round 0.1 · original submission · Minor Revisions

We have received two reviews for your manuscript. Both reviewers found merits in your study and thought it will represent an interesting contribution to the field. However, they also raised a number of minor issues, that deserve further revisions.
In particular, reviewer 1 (F. Huettmann) underlined that data and codes were missing and that a few important ML/AI references were missing as well.
Various other relevant comments regarding the abstract, introduction and discussion were provided by reviewers and should be integrated in the next version (see also annotated ms), as they will help improve the structure and clarity of your manuscript.

·

Basic reporting

Thanks, the MS overall reads well and crisply.

The literature references are too few, and basic ML/AI references are missing.

An ensemble model should refer to J. Friedman, 'many weak learners make for a strong learner'.

Along those lines, the GAM should be dropped, please, because it does not contribute to progress.

I am happy to see that no LM, GLM and AIC, and Bayes were used; that is great.

Text can state what a hypothesis is for them, and how approached here.

Leo Breiman - inference from prediction - should be pursued more.
See paper Breiman 2001 Two Cultures.

All materials and data, and code are NOT well reported, but must be.
GBIF wants a data DOI well shown.

Experimental design

It's a good landscape-scale approach and test, but lacking Alaska and Hawaii, why?

But the metadata is missing to understand the data and code used. It's mandatory.

In the meantime, SDMs do NOT exist as such; they are created and not well defined.
(Maxent, etc), certainly not for ensemble models.

Happy to see no probability was used, but a relative index of occurrence RIO, please say so.

I would have an issue with pseudo-absence (assumed, just a scenario), should be background or real absence, even as a parallel test and scenario.

Validity of the findings

Looks ok to me.

What I am lacking here is a section on ethics:
How about the introduction of exotic species at the cost of others and the wilderness?
That's not what Aldo Leopold would propose.

What about policy, and this work linking with assisting good governance?

The taxonomy of the species should be valid, thus using taxonomic serial numbers TSN from itis.gov.
This does matter for subspecies and genetic setup of the target species; it must be done correctly, and thus use TSN as a basic agreed species description.

Additional comments

Happy to see that LM, GLM, AIC, and p-values are gone; well done. Drop the GAM, please.

Use of GBIF data is great, but then, what about eBird and other sources, open-access data sharing in the wildlife governance discipline, and how to promote it?
That must be stated more clearly.
Thus far, the wildlife and state groups are usually NOT contributing to GBIF much, or open access data sharing; they should, though.

Reviewer 2 ·

Basic reporting

-

Experimental design

-

Validity of the findings

The conclusions drawn correspond to the main idea of the article. They may have important practical significance in the future reintroduction of similar bird species.

Additional comments

The article presented for my review is very well thought out and falls within the scope of the journal. The topic is highly relevant due to the various successes in reintroducing birds into the wild. The models presented have significant practical value and could contribute to greater success in a number of restoration programs for Galliformes birds.

I have some mostly editorial notes, which I have placed as comments in the attached file. I also did not see captions for the figures and tables. I recommended that, for better clarity, the authors add a note to Table 3 explaining what the numbers in the table represent.

Annotated reviews are not available for download in order to protect the identity of reviewers who chose to remain anonymous.

---

## Round 0.2 · accepted · Accept

I am satisfied with the final revisions made to the manuscript.